# Information uncertainty influences learning strategy from sequentially delayed rewards

**Sean R. Maulhardt[1]\*, Alec Solway[1,2], Caroline J. Charpentier[1,2]**

**1** Department of Psychology, University of Maryland College Park, College Park, Maryland, United States of America, **2** Brain and Behavior Institute, University of Maryland College Park, College Park, Maryland, United States of America

\* smaulhar@umd.edu

## Abstract

When receiving a reward after a sequence of multiple events, how do we determine which event caused the reward? This problem, known as temporal credit assignment, can be difficult for humans to solve given the temporal uncertainty in the environment. Research to date has attempted to isolate dimensions of delay and reward during decision-making, but algorithmic solutions to temporal learning problems and the effect of uncertainty on learning remain underexplored. To further our understanding, we adapted a reward learning task that creates a temporal credit assignment problem by combining sequentially delayed rewards, intervening events, and varying uncertainty via the amount of information presented during feedback. Using computational modeling, two learning strategies were developed: an eligibility trace, whereby previously selected actions are updated as a function of the temporal sequence, and a tabular update, whereby only systematically related past actions (rather than unrelated intervening events) are updated. We hypothesized that reduced information uncertainty would correlate with increased use of the tabular strategy, given the model's capacity to incorporate additional feedback information. Both models effectively learned the task, and predicted choices made by participants (N = 142) as well as specific behavioral signatures of credit assignment. Consistent with our hypothesis, the tabular model outperformed the eligibility model under low information uncertainty, as evidenced by more accurate predictions of participants' behavior and an increase in tabular weight. These findings provide new insights into the mechanisms implemented by humans to solve temporal credit assignment and adapt their strategy in varying environments.

## Author summary

People routinely experience uncertain and temporally extended environments that might be encountered again in the future. To overcome the burden of

---

**Data availability statement:** Data and code can be accessed at https://osf.io/rp56b/.

**Funding:** This work was supported by the University of Maryland start-up funds to A.S and C.J.C. CJC is also supported by a National Institute of Mental Health R00 award (R00MH123669). The funders had no role in study design, data collection and analysis, decision to publish, or preparation of the manuscript.

**Competing interests:** The authors have declared that no competing interests exist.

relearning these environments, people detect patterns that will facilitate future decision-making. Although pattern detection is complex, one such mechanism – known as credit assignment – is particularly important when identifying decisions to repeat and those to avoid. Credit assignment allows people to assign value to past experiences and then utilize these values for quick and efficient decision-making. However, this process becomes substantially more complex when the delay between a decision and its associated outcome increases, and when the available information decreases, making the environment more uncertain. Our research experimentally isolates the effects of delay to understand the strategies people use to solve credit assignment problems in environments with low and high uncertainty. We discover that information uncertainty influences how credit assignment strategies are deployed. Our methods, analyses, and results pave the way for more complex real-world experimentation that seeks to understand the interaction between reward, time, and uncertainty.

## Introduction

The temporal credit assignment (CA) problem emerges when an agent receives a reward and must determine which prior event caused that specific outcome. Commonly, mechanisms of CA occur throughout our daily lives so that we might learn to credit the best course of action in pursuit of maximizing future reward. For example, if you receive a compliment on a recent dinner meal, then you must recall past behaviors that contributed to the overall pleasantness of the meal. While a pinpointing comment (e.g., "the seasoning was perfect") provides clear guidance, general praise (e.g., "great dinner!") introduces uncertainty about which elements truly enhanced the dining experience. Questions may arise, such as "Was the entire dinner enjoyable, or just certain parts of it?" or "Which actions should be replicated in future meals?" To mitigate the challenges of CA, individuals often deploy strategies that will reduce uncertainty about which actions led to success, such as using a predetermined recipe or a prepackaged dinner. The complexity of any CA evaluation relies on the order of event onset, interval between events, information available, state and temporal uncertainty, or availability of resources [1–3].

The complexity of the temporal CA problem is defined by Minsky [4] who characterizes the difficulty for machines to ascribe relationships in sequential, probabilistic, and/or obscured informational environments. Humans solve this problem using varying amounts of information - a process that has become the focus of experiments seeking to understand how the brain implements these algorithms. The Rescorla-Wagner model characterizes how expectation errors drive learning in simple conditioning, where actions and outcomes have clear and immediate relationships [5–7]. Building upon this framework of prediction errors led to additional learning mechanisms that could be sensitive to the sequential nature of the environment. The temporal difference (TD) learning model allows the chaining of reinforcement signals to past actions and thus solves credit assignment through the recency

of experienced states [8–10]. Finally, hybrid models that combine TD learning with a model-based transition matrix are thought to overcome CA problems that require the estimation of state uncertainty [11–15]. While delay, or temporal uncertainty, naturally emerges in these sequential designs, the primary focus of investigation leans more towards the structure of state relationships than the sequence of temporal relationships [16–18]. Consequently, we lack a clear understanding of how humans implement different algorithmic solutions under temporal uncertainty, and how environmental factors influence their strategy selection.

While various algorithmic solutions exist for credit assignment under temporal uncertainty, their empirical validation in human learning remains limited [19,20]. Previous work has examined related phenomena: the effect of delay on memory [21], temporal preferences for rewards [22,23], and the subjective experience of waiting times [24]. Furthermore, most of these experiments on delay occurred in participants who had access to full information. However, isolating the specific effects of delay on human reinforcement learning (RL) has proven challenging for several reasons [18,25]. First, delay inherently introduces uncertainty in both the temporal sequence of events and the assignment of credit. Second, as temporal delays increase, algorithmic solutions become progressively less efficient, making it difficult to distinguish learning deficits from computational constraints. Third, longer or varying delays expand the possible solution space, vastly increasing the array of individual learning strategies that may emerge.

The possible individual solutions to the problem of reward evaluation under temporal uncertainty often turn towards two predominant classes of RL algorithms – retrospection and prospection [19,20,26]. Retrospective CA, such as via an eligibility trace mechanism, evaluates past events to determine which are eligible to be assigned credit based on their temporal sequence and relationship to the outcome. However, when intervening events disrupt the temporal continuity between action-and-reward pairs, an agent may erroneously assign credit by naively following the sequence of transitions. To overcome some of these uncertainties, prospective methods attempt to predict and account for future temporal relationships. An agent prospects about future reward expectations and increases attentiveness at critical times when reward should be delivered [27]. Critically, both methods rely on the temporal horizon and observability of the state space, but prospective methods require an accurate belief state of the environment to perform planning behavior [28,29].

Recent efforts in RL have identified that when multiple algorithmic solutions to a problem are available, agents are likely to arbitrate over time and assign control over behavior to the most reliable strategy given the environment [13,14,30,31]. To validate this framework, one should establish clear experimental manipulations that would shift prospective and retrospective solutions, such as varying the degree of information uncertainty. This leads to our final questions: whether reward learning under temporal uncertainty can be empirically solved by a mixture of prospective and retrospective strategies, and whether we can experimentally shift control to the most reliable strategy by manipulating information uncertainty through the observability of the reward function [32]. Our hypothesis posits that learning performance in a temporal learning task can be best explained by a mixture of retrospective and prospective algorithms, and that less observable environments will promote the shift to more retrospective algorithms.

## Results

Participants (N = 142) completed two separate sessions of a learning task that required individuals to solve a temporal credit assignment problem under low and high information uncertainty (Fig 1, see Methods for full details). Each session of the learning task contained 8 stimuli; four of which were associated with a reward delivered immediately, and the other four had delay contingencies in which their rewards were revealed two trials into the future. All stimuli were paired up equally for a total of 336 trials, and on each trial, participants chose between a randomly presented pair of stimuli and observed the reward (Fig 1A-1B). Additionally, stimuli were associated with differing starting rewards (4, 1, -1, -4) that slowly changed over time (Fig 1C). The main experimental manipulation, implemented across sessions, modulated information uncertainty based on how reward was revealed: in the conjoint condition, reward was presented as the summed total of immediate and delayed reward (Fig 1A), while in the disjoint condition, information about immediate and delayed

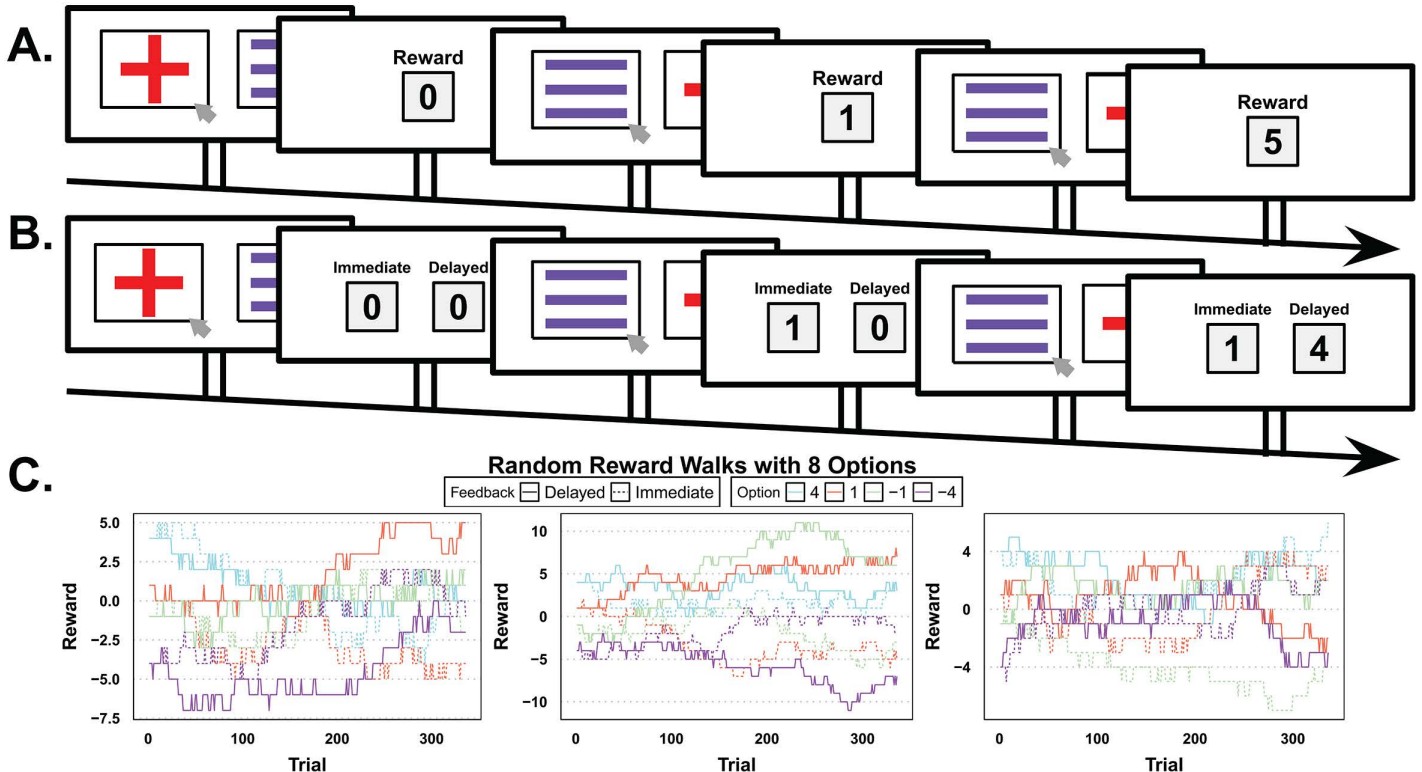

**Fig 1. Experimental design and random walks. A.** Example trial sequence of the conjoint condition. Participants choose between two objects, then receive feedback. Here, a purple three-line offers an immediate reward of '1', while a red cross provides a delayed reward of '4' after two trials. **B.** The disjoint condition is presented with the same sequence of events but with dissociable feedback. **C.** This illustrates the three fixed random walks reward value patterns for all eight stimuli across trials. Each stimulus is linked to either an immediate (solid lines) or a delayed (dashed lines) reward. The starting reward values are 4, 1, -1, or -4, and they gradually drift throughout the task. Participants were randomly assigned to one of the three random walks at the beginning of each session (hence could experience different random walks in conjoint and disjoint condition) and encountered a random sequence of unique pairs (28 in total) in each session. Schematic elements designed in Google Slides per Google's copyright agreement.

rewards was presented separately (Fig 1B). Condition order (i.e., which session was completed first) was counterbalanced across participants, yielding two groups: disjoint first (N = 74) and conjoint first (N = 68).

## Decisions favor optimality in the first disjoint stage

Initial behavioral analysis sought to uncover participants' choice behavior from decision metrics, specifically optimal choice, and average outcome received per trial (Table 1). Optimal choice was calculated as the proportion of trials where the higher reward object was selected. The average outcome was the reward amount associated with the selected option, averaged across all trials. Noteworthily, a participant choosing the optimal option more often than chance (50%) reflects learning performance about the delayed contingencies and random walk of reward. Optimal choice was placed into a one-sample t-test relative to chance ($\mu = .5$) for each of the four combinations of reward condition and stage (see all means and standard deviations in Table 1), resulting in performance for all conditions being significantly above chance, $t(73) = 15.92$, $p < .001$. To quantify how performance varied across conditions, each variable was then placed into a linear regression with an interaction term between reward condition (disjoint, conjoint) and stage (first, second). Residual and QQ plots were all reasonable and did not merit the use of nonparametric tests. For optimal choice, there was a significant main effect of reward condition, $b = .1$, $SE = .018$, $t(280) = 5.32$, $p < .001$, $\omega_p^2 = .09$, a significant effect of stage, $b = .04$, $SE = .018$, $t(280) = 2.02$, $p = .04$,

**Table 1. Descriptive statistics for behavioral performance.**

| Disjoint-1 | Mean | SD | Disjoint-2 | Mean | SD |
|---|---|---|---|---|---|
| Outcome | 0.85 | 3.18 | Outcome | 0.61 | 3.25 |
| Optimal | 0.71 | 0.45 | Optimal | 0.64 | 0.48 |
| **Conjoint-1** | **Mean** | **SD** | **Conjoint -2** | **Mean** | **SD** |
| Outcome | 0.48 | 3.17 | Outcome | 0.66 | 3.12 |
| Optimal | 0.61 | 0.49 | Optimal | 0.65 | 0.48 |

Means and standard deviations (SD) are shown for outcome received (Outcome), along with proportion of choosing the optimal option (Optimal) across the two reward conditions (Disjoint and Conjoint) and current stage (1 and 2). Note that disjoint-1 and conjoint-2 represent the same group of participants (N = 74) while conjoint-1 and disjoint-2 represent another group (N = 68).

$\omega_p^2 = .01$, and a significant interaction, $b = -.1$, SE = .025, $t(280)$ = -4.05, $p < .001$, $\omega_p^2 = .05$. For average outcome, there was a significant main effect of reward condition, $b = .36$, SE = .15, $t(280)$ = 2.47, $p = .01$, $\omega_p^2 = .02$, no effect of stage, $p = .23$, and a significant interaction, $b = -.42$, SE = .21, $t(280)$ = -2.00, $p = .047$, $\omega_p^2 = .01$. As expected, the main effects of reward conditions suggest that performance and outcomes were both higher in the disjoint compared to conjoint condition. Additionally, the interactions between the stages and reward conditions indicate that starting in the disjoint condition resulted in higher performance and outcomes (Table 1). Participants performed best when starting in the disjoint condition, but they also performed in the conjoint condition comparably to those who ended in the disjoint condition. Thus, the group that started in disjoint and ended in conjoint performed better than the group that started in conjoint and ended in disjoint.

## Computational models predict behavioral signatures of learning

Two competing RL models were implemented as potential solutions to the credit assignment problem in our task (see Methods for details and equations). The first RL model was an eligibility trace (abbreviated 'Elg'), a retrospective solution which naively credits the temporal sequence of previous choices and contains three specific parameters, learning rate (alpha), credit decay (lambda), and SoftMax inverse temperature (beta). The second RL model was our tabular model (abbreviated 'Tab'), a prospective solution which systematically credits the appropriate trials in the temporal sequence (0 and 2F) and contained three of the same parameters. We provide a graphical representation in Fig 2 and recreate the credit assignment mechanisms from our data (S1 Fig).

To validate our eligibility and tabular models and to understand their relation to behavioral data along with the behavioral patterns they each capture, we defined behavioral measures that show whether participants were tracking delayed rewards correctly. Initially, we expected certain sequence patterns to reveal delayed reward learning through participants' stay or switch behavior, akin to the key behavioral signature of the two-step task [11]. Specifically, we selected sequences where participants chose a delayed option, and it reappeared as a potential choice three trials later. Overall, about 15.7% of the original dataset was subset across individuals and showed a range of 10.5% to 22.8%. We then analyzed their decision to stay on their delayed option choice three trials in the future using the following multilevel logistic regression (equation 1).

$$Stay \sim Condition * Time * Reward + (1 + Condition * Time * Reward \mid Subject) \tag{1}$$

Specifically, the regression examined whether the experimental conditions (*Condition: conjoint, disjoint*) influenced participants' decision to stay (*Stay: 1 stay, 0 switch*) while accounting for the reward valence (*Reward: + positive feedback, - negative feedback*) received from current and two-trials forward (*Time: 0F, 2F*). For example, a participant might erroneously stay on their choice three trials in the future due to a positive immediate reward but should have correctly switched due to

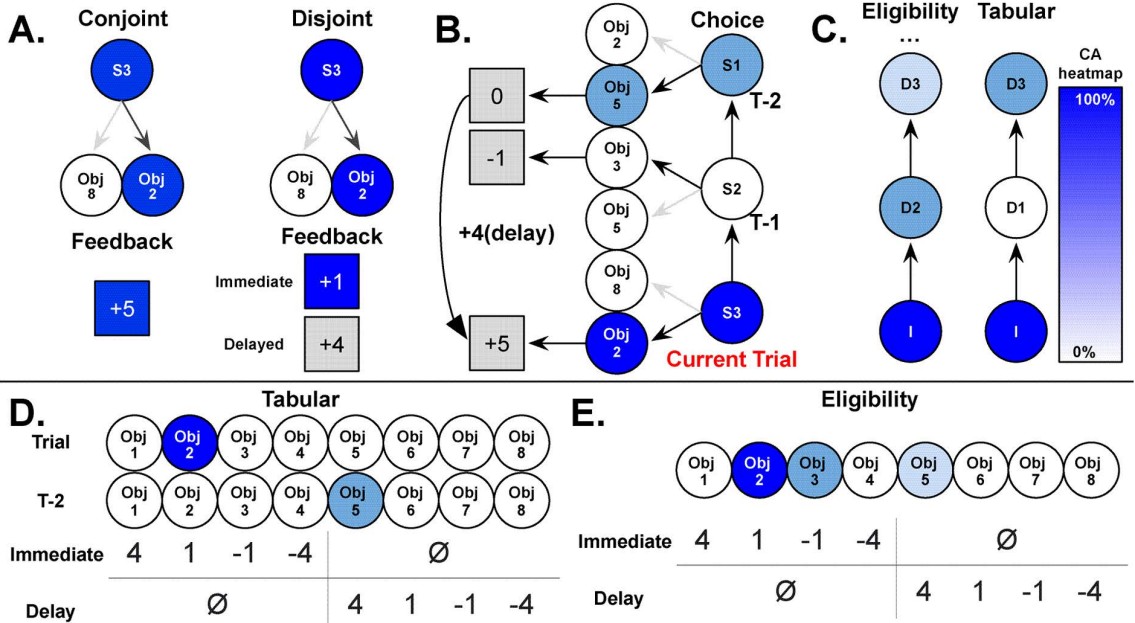

**Fig 2. Graphical representation of task conditions, learning models, and value functions. A.** Differences in feedback presentation based on the participant's condition and the outcome used to generate the prediction error for immediate feedback. **B.** Example sequence of two-alternative forced choice trials and the participant's selection (darker arrow) of object 2 (Obj 2) in the current trial (red), the previous trial (T-1), and the trial-minus-two (T-2). The colors correspond to the tabular model, which updates the immediate choice (+1) and trial-minus-two choice (+4), each generating a prediction error to update the value function. **C.** Temporal sequence of assigning credit (shown as a blue heatmap). In this model, the tabular model skips assigning credit to the previous state (S2). The triple period signifies that credit assignment can extend beyond the three depicted states. Note that the extent to which past states are assigned credit in each model depends on the free parameter lambda: for eligibility, higher lambda values mean that credit extends further back in time (less decay), while specifically for tabular, higher lambda values mean less discounting of the trial-minus-two state. **D-E.** Value functions for the tabular model **(D)**, which involves separate, independent, updates for the immediate and delayed chosen options, and for the eligibility trace **(E)**, which utilizes a single prediction error for updates. S: State, Obj: Object, I: Immediate, D: Delay.

a negative reward two-trials in the future (Fig 3A). Using equation 1, we conducted three logistic regression analyses: one with the participants' data, and two with a simulated dataset using 25 iterations of each participant's best-fitting parameters (see Methods for details) for both the eligibility and tabular choice models (Fig 3B).

The three-way interaction (disjoint $*+*$ 2F) was significant for all three logistic regression models (Fig 3B), odds-ratios (OR) for participant, OR =1.97, SE = 1.186, Z = 4.19, p < .001, 95% Confidence Interval (CI) [1.44,2.71]; for eligibility, OR =1.53, SE = 1.18, Z = 2.6, p = .01, 95% CI [1.02, 1.32]; for tabular, OR=1,65, SE = 1.17, Z = 3.23, p = .001, 95% CI [1.22, 2.23]. This shows that our three-way interaction was justified for our participants and our model. For participants' data (Fig 3B – red line), the interaction was such that in the conjoint condition, the reward valence difference on choice (probability of stay difference of positive minus negative rewards denoted $\Delta_{+-}$) was similar at both times ($\Delta_{+-}$0F 95% CI [.07,.16], $\Delta_{+-}$2F 95% [.09,.16]). However, in the disjoint condition, the effect of reward valence on choice was strongest for two-trials forward ($\Delta_{+-}$0F 95% CI [.04,.13], $\Delta_{+-}$2F 95% [.21,.3]). Given this effect, we show that participants are indeed using the correct reward information and that our manipulation was effective.

To better understand how our models mapped with participants' behavior, we inspected the extent to which the 95% CI overlapped between the model predictions and the participants' data. Fig 3B shows margin overlaps with text annotations, while differences between positive and negative rewards are reported with 95% CIs in the text. The three-way interaction shows that in the conjoint condition, tabular ($\Delta_{+-}$0F 95% CI [.05,.12], $\Delta_{+-}$2F 95% CI [.11,.17]), and eligibility ($\Delta_{+-}$0F 95% CI [.06,.14], $\Delta_{+-}$2F 95% CI [.02,.09]) models were similar and both overlapping with participants' means ($\Delta_{+-}$0F 95% CI [.07,.16],

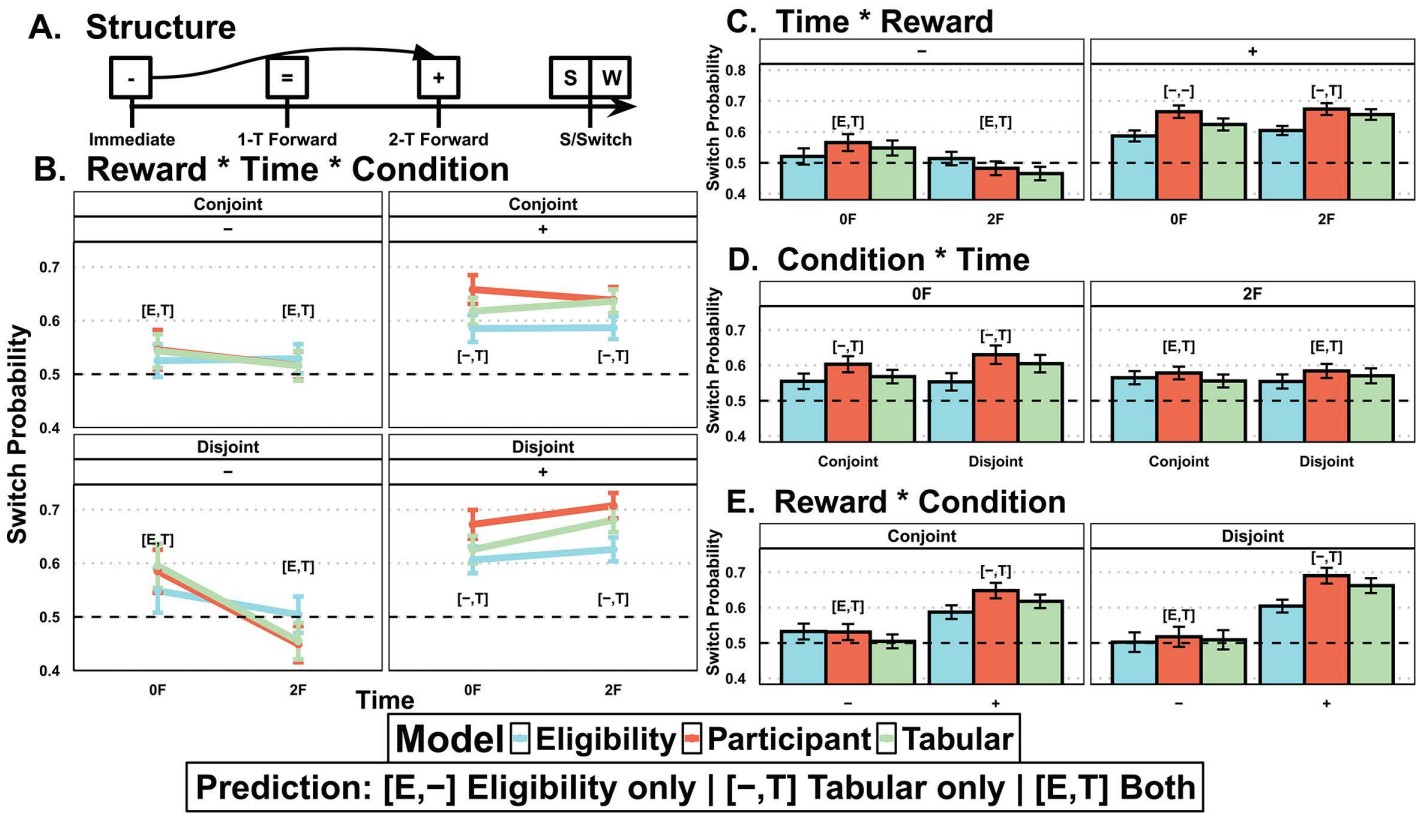

**Fig 3. Behavioral signatures of learning and model predictions. (A)** This diagram demonstrates the logical structure of a delayed choice that reappears after three trials, categorized as either 'stay' (S) or 'switch' (W). It includes feedback valence for immediate and two-trials forward choices. Correctly utilizing two-trials forward information (illustrated with a curved arrow) suggests staying with the initial choice. **(B)** The probability of maintaining (i.e., staying on) a delayed choice three trials later was modeled using multilevel logistic regression, as a function of condition (conjoint or disjoint), time (immediate, 0F or two-trial forward, 2F), and reward (positive, + or negative, -), as well as their interactions (equation 1). This regression was run on data from participants (red), as well as data generated by the tabular model (green) and data generated by the eligibility model (blue) for comparison. The logistic regression's estimated marginal means are displayed, accompanied by 95% confidence intervals for each condition. Text displays when participants' estimated confidence intervals overlap with eligibility [E,-], tabular [-,T], or both [E,T]. **(C-E)** Collapsed two-way interactions for each of the combinations between the three variables, specifically time*reward collapsed across condition **(C)**, condition*time collapsed across reward **(D)**, and reward*condition collapsed across time **(E)**. Bars represent marginal means; error bars represent 95% confidence intervals.

$\Delta_{+-}$2F 95% [.09,.16]). However, in the disjoint condition, the tabular model mapped to participants' data better than eligibility for 2F ($\Delta_{+-}$2F 95% CI tabular [.2,.28], participants [.21,.3], eligibility [.08,.16]), with both overlapping for 0F ($\Delta_{+-}$0F 95% CI tabular [.02,.11], participants [.04,.13], eligibility [.01,.1]).

The two-way interactions helped to identify the dimensions that were most prevalent for identifying differences in participant choice between the two models. Although some of these effects were significant, we were more concerned about how well the model tracked win-stay, lose-shift behavior in the participants. Collapsing across conditions, we found that positive rewards (+) showed a stronger win-stay than negative rewards (-) resulting in a lose-shift heuristic (Fig 3C), which was predominately captured in the tabular model for delayed options (2F). When we collapse on reward, we see that tabular uniquely tracked participants' patterns in immediate (0F) but both models tracked participants in delayed (2F) (Fig 3D). Lastly, collapsing across time showed that only tabular tracked participants for positive reward (+) but both models tracked negative reward (-) effects (Fig 3E). Altogether, we find evidence for both models picking up unique patterns that might justify the use of a hybrid model.

## Tabular and eligibility predictions best map onto disjoint and conjoint conditions, respectively

In addition to the independent tabular and eligibility models, a hybrid model combining the two strategies was also defined, whereby the independent beta parameter for each model determined the contribution of that strategy in the combined SoftMax equation (see Methods for details and equations). Our second posterior predictive check analysis was similar to Tanaka et al. [26], which initially focused on illustrating how participants were capable of tracking rewards in specific example pairs. We first showed participants' ability to track reward in some example pairs (Fig 4A), as well as how our hybrid model mimics this choice trajectory (Fig 4B). Note that these example pairs were chosen to show a clear reversal, but given the nature of the random walks, some other pairs exhibited multiple reversals, and some exhibited none. Next, we set out to see if our models behaved systematically different across the different pair types and utilized our 25-iterations of model choice using all 142 participants' pairs and reward sequences. For each of the three models

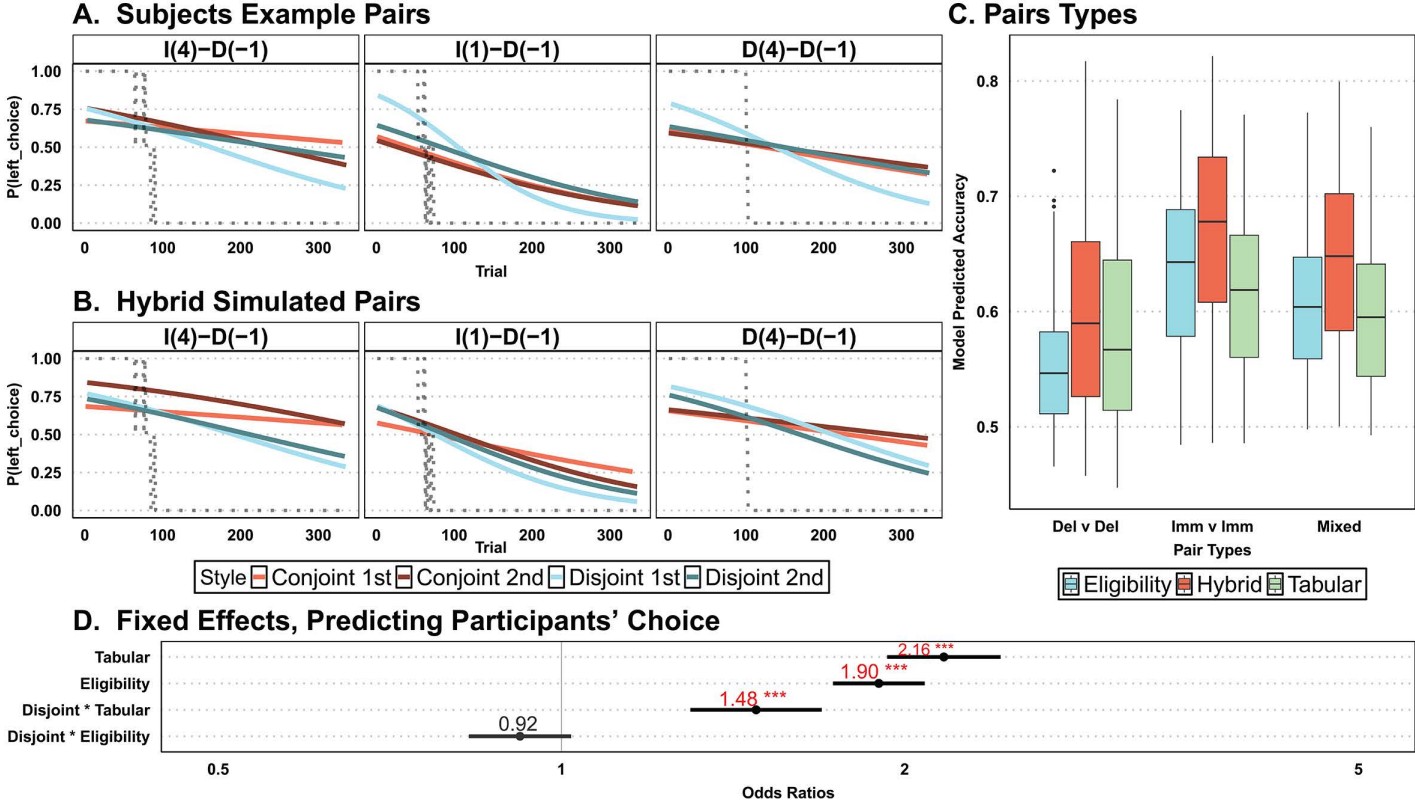

**Fig 4. Model prediction of trial-by-trial choice. (A)** Average participants' propensity to choose left on three example pairs from one of the reward random walks, selected to show a clean reversal moving from left-to-right choice. The title identifies the reward contingency (I = Immediate, D = Delayed) with the initial value enclosed in parentheses (4 = initial start 4, -1 = initial start -1). The y-axis is the probability of selecting the left choice from the title, such as I(4)-D(1) illustrating the left choice to be the immediate option with starting value 4. The grey dotted line tracks when the optimality of the reward shifts from left (1.00) to equivalent (0.50) and then to right (0.00). Thus, participants in the different conditions (disjoint, conjoint) and different stages (1st, 2nd) should follow the optimality of reward with participant data on top. Further, lighter colors belong to the same group as darker colors. **(B)** The same analysis was performed on choice data generated by the hybrid model, using the best-fitting participant parameters. **(C)** Predictive accuracy (i.e., percentage of model-predicted choice matching participant choice) for each model and each pair type with delayed vs. delayed (Del v Del), immediate vs. immediate (Imm v Imm), and immediate vs. delayed (Mixed), shown as a boxplot across participants' best-fitting parameters. **(D)** Fixed effects from a logistic regression predicting choice on each trial from tabular predictions, eligibility predictions, and their interaction with condition (equation 2), aligning with our hypothesis that tabular predicts behavior better in disjoint than in the conjoint condition (significant disjoint * tabular interaction), while eligibility only showed the opposite direction (non-significant disjoint * eligibility interaction). Dots and associated numbers represent the odds ratio for each effect; horizontal error bars represent 95% confidence intervals. *** *p* < .001.

we calculated the average model accuracy (i.e., how well the model predicted participants' choices) separately for each pair type. We show that the tabular model appeared to perform better than the eligibility model in delay vs. delay pairs, and worse than eligibility for immediate vs. immediate pairs (Fig 4C). The hybrid model performed better for all pair types (Fig 4C, red bars), suggesting that given its six parameters, it may find parametric spaces that accurately map immediate, delayed, and mixed pair types without an overt bias towards one or the other.

For a comprehensive understanding of overall trial-by-trial accuracy and decision-making patterns in our two models (tabular and eligibility) under both conditions (disjoint and conjoint), we expanded our analysis. To quantify the models' (eligibility and tabular) trial-by-trial predictive accuracy and to compare it between conditions, we ran another multilevel logistic regression (equation 2), collapsing across all random walks and pairs.

$$Choice \sim Condition * (Elg + Tab) + (1 + Condition * (Elg + Tab) \mid Subject) \tag{2}$$

We were particularly interested in understanding how the predictions made by the tabular and eligibility models corresponded to disjoint and conjoint conditions, respectively. Specifically, we anticipated that the tabular model would predict participants' choices more accurately in the disjoint condition and the eligibility model would predict participants' choices more accurately in the conjoint condition. To test this, we employed the same task sequences observed by participants and utilized a beta distribution ($\alpha = 1$, $\beta = 2$) for our learning rate, a gamma distribution ($\alpha = 3$, $\theta = 1$) for the inverse temperature parameter, and a beta distribution ($\alpha = 2$, $\beta = 3$) for the decay parameter. This maintained consistent parameters across the two models and kept a high inverse temperature parameter to reduce random noise. This also allowed us to generate choice probabilities for both models to see which were better predictors of participants' choice under the differing conditions (equation 2, Fig 4D). Using the 'mixed' function in 'afex', which prevents inflated Type I errors [33], we found a positive interaction between tabular predictions and condition, OR = 1.48, $\chi^2(1) = 30.14$, $p < .001$, such that the predictions of the tabular model explained participants' choices better in the disjoint compared to the conjoint condition, as hypothesized. For the interaction between eligibility predictions and condition, we found a numerical reduction in the odds-ratio, but this effect was not significant, OR = .92, $\chi^2(1) = 2.46$, $p = .12$.

## Best-fitting model varies by condition

After validating our models, we proceeded to fit them to the data and extract model-fitting metrics and parameter values (see Methods for details about the model-fitting procedure). We provided two descriptive statistics of model fit averaged across participants for each condition (Table 2): Akaike Information Criterion (AIC) and pseudo-$R^2$. In all cases, the hybrid model showed a better fit to participant data but could not always be significantly differentiated from the eligibility model ($p > .05$). To note, the hybrid model can be reduced to either the eligibility or tabular model but will still be penalized for the additional parameters. The tabular model only appeared to fit better in the Disjoint-1 condition when compared to eligibility model in all other conditions. Consistent with our posterior predictive checks from Fig 4D, the eligibility model fit did not differ across conditions, while the tabular model fit improved in the disjoint condition (compared to the conjoint condition). Participants who started in the disjoint condition had better model fits than those who started in the conjoint condition, which aligns with our previous descriptive statistics (Table 1).

## Tabular model weight increases in the disjoint condition

Next, each of the RL model parameters was used in a combination of various statistical tests to assess whether each parameter differs between strategies (eligibility vs. tabular) and between conditions (conjoint vs. disjoint) (Table 3), as well as whether these effects of condition interacted with phase order (i.e., which condition was completed first) (Fig 5). In particular, the two strategy weights (beta parameters), measuring the contribution of our strategies, were used to validate our hypothesis that low information uncertainty maps to a prospective solution (tabular) and high information uncertainty maps to a retrospective solution (eligibility).

**Table 2. Average model-fitting metrics.**

| | Group: Disjoint→Conjoint (N=74) | | | | | | Group: Conjoint→Disjoint (N=68) | | | | | |
| | Disjoint-1 | | | Conjoint-2 | | | Conjoint-1 | | | Disjoint-2 | | |
| | AIC | p | $R^2_p$ | AIC | p | $R^2_p$ | AIC | p | $R^2_p$ | AIC | p | $R^2_p$ |
|---|---|---|---|---|---|---|---|---|---|---|---|---|
| Elg | 360 | .001 | .24 | **361** | **.78** | .24 | **392** | **.08** | .17 | **391** | **.29** | .17 |
| Tab | 358 | .001 | .25 | 383 | .001 | .19 | 411 | .001 | .13 | 400 | .001 | .15 |
| Hybrid | **343** | – | .29 | **358** | – | .26 | **391** | – | .19 | **384** | – | .2 |

Average model-fitting metrics (Akaike information criterion, AIC) for each model across condition and stage, with lower AIC values representing better fit. P-values were calculated using paired Wilcoxon signed-rank tests compared to the best-fitting (indicated by -) model in each condition and highlighted in bold to indicate when a model couldn't be statistically separated. Pseudo-$R^2$ (denoted $R^2_p$) represents the proportion of variance accounted for in participants' choices relative to a random model with higher values representing better fit. $R^2_p$ was not used for model comparison and is provided for general intuition of model accounted variance.

We found that the tabular beta parameter, but not the eligibility beta parameter, varied with condition. Specifically, the weight of the tabular model was larger in the disjoint relative to the conjoint condition, consistent with our hypothesis of increased reliance on tabular in the disjoint condition and the model fits from Table 2 above. Interestingly, the eligibility decay rate also varied with condition, with a less distant trace in the conjoint condition. Finally, the learning rate in the tabular model appeared to have larger updates of the prediction error in the disjoint condition, but one should be mindful in interpreting this effect given the recovery rate of this parameter (S1 Text). All t-tests are shown in Table 3.

To assess whether these effects of condition held across stage order groups and to examine possible order effects of completing one or the other condition first, we then ran mixed-effects linear regressions predicting each parameter value from condition, stage order group, and their interaction.

$$Parameter \sim Condition * Stage + (1 \mid Subject) \tag{3}$$

For beta-tabular, an unconditional model allowed an estimate of the intraclass correlation (ICC, see Methods for details), accounting for 39.26% of the total variation between participants. The best fitting model was one without an interaction term, showing a marginally significant change in deviance compared to the model without an interaction, $\chi^2(1) = 3.74$, $p = .053$. The final non-interaction model showed a significant main effect of condition, $b = .12$, $SE = .03$, $\chi^2(1) = 15.55$, $p < .001$, and a significant main effect of stage order group, $b = -.16$, $SE = .04$, $\chi^2(1) = 12.56$, $p < .001$. The effect was such that beta-tabular was higher in the disjoint than the conjoint condition, as predicted, but there was an additional effect of

**Table 3. Differences in model parameters by condition, for each strategy.**

| Variable | df | t | p | 95% CI | d |
|---|---|---|---|---|---|
| AlphaElg | 141 | 1.26 | .211 | [-.02,.08] | .11 |
| AlphaTab | 141 | -4.32 | **<.001**** | [-.21, -.08] | -.36 |
| BetaElg | 141 | .28 | .78 | [-.08,.1] | .02 |
| BetaTab | 141 | -4.04 | **<.001***** | [-.18, -.06] | -.34 |
| LambdaElg | 141 | -3.63 | **<.001**** | [-.18, -.05] | -.3 |
| LambdaTab | 141 | .14 | .89 | [-.06,.07] | .01 |

Two-tailed paired t-tests were computed for individual parameters extracted from each independent model (Elg: Eligibility; or Tab: Tabular): learning rate (Alpha), decision weight (Beta) and decay rate (Lambda), comparing conjoint versus disjoint condition. df, degrees of freedom associated with each t-test; t, t-statistic, with negative numbers representing a larger mean in disjoint than conjoint condition; p, p-value, * p<.05 ** p<.01 *** p<.001, 95% CI, 95% confidence interval; d, effect size calculated as Cohen's d.

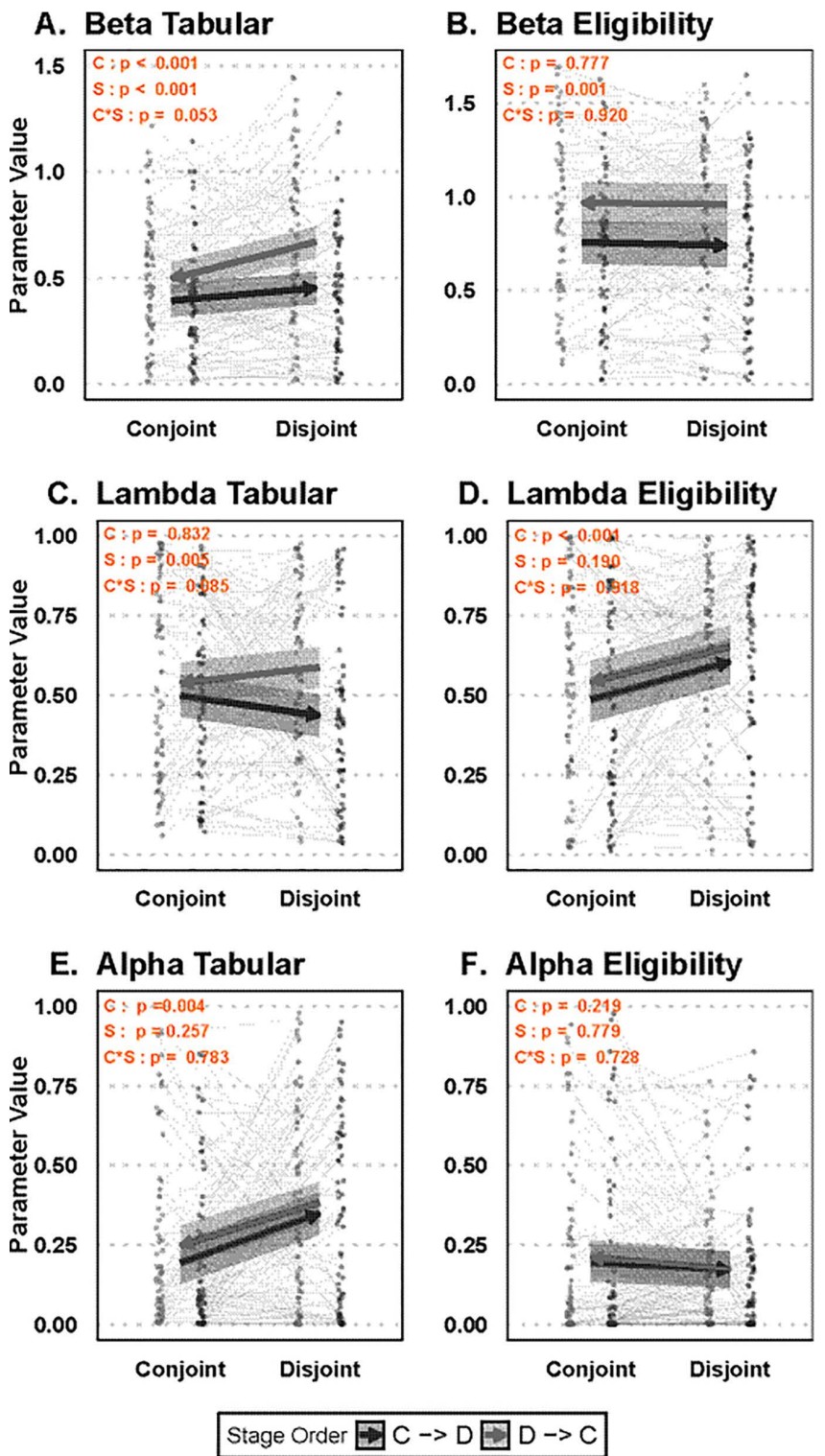

**Fig 5. Effect of condition and stage order group on computational model parameters.** A mixed-effects linear regression model was applied to each RL parameter from their yoked independent model, predicting the parameter value from condition, stage order group, and their interaction (equation 3). Specifically, this regression was performed on the decision weight - Beta - for both tabular **(A)** and eligibility **(B)**, decay rate - Lambda - for tabular **(C)**, eligibility **(D)**, as well as the learning rate - Alpha - across tabular **(E)** and eligibility **(F)**. The results display estimated marginal means, marked with

arrows pointing towards the participants' final condition (Conjoint and Disjoint) and the two groups based on the order of stages, namely from conjoint stage 1 to disjoint stage 2 (C→D, black) and from disjoint stage 1 to conjoint stage 2 (D→C, grey). Shaded areas represent 95% confidence intervals. Additionally, individual dots and lines show parameter estimates for each participant. The top left corner of the results highlights the p-value associated with the main effect of the condition (C), the main effect of the stage (S), or their interaction (C * S).

stage order such that starting in disjoint condition led to greater values of beta-tabular than starting in the conjoint condition (Fig 5A), consistent with the higher performance and higher tabular model fits in that group. However, slope differences between stage order groups had only marginal effects.

For beta-eligibility, an unconditional model allowed an estimate of the ICC, accounting for 36.38% of the total variation between participants. The best fitting model held without an interaction term, showing a non-significant change in deviance when comparing models with and without the interaction term, $\chi^2(1) = .01$, $p = .92$. The final model showed no effect of condition, $p = .78$, but a significant effect of stage order, $b = .21$, SE $= .07$, $\chi^2(1) = 10.61$, $p = .001$. The main effect of stage order showed overall higher beta-eligibility in participants who started in the disjoint condition compared to those who started in the conjoint condition (Fig 5B).

For lambda-tabular, an unconditional model allowed an estimate of the ICC, accounting for 6.15% of the total variation between participants. The best fitting model held without an interaction term, showing a marginally significant change in deviance compared to the model without the interaction, $\chi^2(1) = 3.02$, $p = .08$. There was no significant effect of condition, $p = .83$, but there was an effect of stage order, $b = .1$, SE $= .03$, $\chi^2(1) = 7.95$, $p = .005$ (Fig 5C). In effect, there was a decrease in weight of the tabular model when moving to the second stage, regardless of the condition.

For lambda-eligibility, an unconditional model allowed an estimate of the ICC, accounting for 13.68% of the total variation between participants. The best fitting model consisted of a non-interaction model as the change in model deviance was not significant, $p = .92$. The non-interaction model showed a significant effect of condition, $b = .11$, SE $= .03$, $\chi^2(1) = 12.66$, $p < .001$, and a non-significant effect of stage order, $p = .19$ (Fig 5D). Participants in the disjoint condition appeared to have lower discounting rates for delayed updates compared to the conjoint condition, indicating that the eligibility trace decayed at a slower rate.

For alpha-tabular, the fit was based on a generalized least squares model with compound symmetry, as the within-subject correlation was negative, $\rho = -.05$. The negative correlation indicates that the within-subject effect appears to be inversely correlated, such that when a person has a higher learning rate, they are more likely to have a lower learning rate in the next stage or vice-versa. Maximum likelihood was used to compare models, which did not support adding the interaction term, likelihood ratio test, $p = .78$. The non-interaction model showed a significant effect of condition, $b = .15$, SE $= .03$, $\chi^2(1) = 18.65$, $p < .001$, and a non-significant effect of stage order, $p = .18$ (Fig 5E). Generally, the learning rate was higher for participants in the disjoint condition compared to the conjoint condition.

For alpha-eligibility, an unconditional model allowed an estimate of the ICC, accounting for 29.93% of the total variation between participants. The best fitting model consisted of a non-interaction model as the change in model deviance was not significant, $p = .73$. The non-interaction model showed no significant effects of condition, $p = .21$, or stage order, $p = .78$ (Fig 5F).

## Discussion

When presented with sequentially delayed rewards, the problem of credit assignment (CA) requires a person to engage in an intuitive or strategic solution [4]. This question of individual solutions was posed using a novel reward learning task under the guise of two differing strategies of temporal CA [26,29]. We also questioned whether changes in strategy within individuals would be based on the uncertainty in the environment. To achieve this objective, we manipulated the degree of information in feedback presentation, which modulated information uncertainty through a partially and fully observable reward function. The eligibility trace, a viable and versatile retrospective solution for unobservable environments, was

contrasted with a tabular model which differentiated credit assignment on the dimension of time in a more prospective manner [10,26]. The eligibility trace updates with a single prediction error that decays the reinforcement signal over the sequence of past actions. In contrast, the tabular model employs two distinct prediction errors along the immediate and two-trial back timing but collapses across the time dimension rather than differentiating actions. Predictively, the learning efficiency differs between the two strategies but either strategy can fully observe the reward function over time. In contrast to previous work [26], we also implemented random reward walks to prevent full learning. We predicted that in the disjoint condition, which made immediate versus delayed reward information available, the tabular strategy would have greater utilization rates to make use of that information, while in the conjoint condition, where no such detailed information was provided, the eligibility trace might be defaulted to.

Our findings were consistent with this prediction across multiple analyses. First, the tabular model was found to capture clear patterns of behavior, specifically the tendency to repeat the choice of a delayed reward option in the disjoint versus conjoint condition, which the eligibility model did not capture. Second, the tabular model fits were greater in the disjoint condition than in the conjoint condition, while the eligibility model fits remained stable across conditions. Third, predictions of the tabular model explained participants' choices better than predictions of the eligibility model in the disjoint condition, while the opposite was found in the conjoint condition. Interestingly, better performance of the eligibility trace model in the conjoint condition replicated the results of Tanaka et al. [26]. Finally, parameters from the tabular and eligibility models provided more insights into the specific mechanisms deployed by participants to adapt to the change in information uncertainty about delayed rewards. Regarding these mechanisms, the strategy weight of the contribution from the tabular model was overall higher during the disjoint condition compared to the conjoint condition with an additional increase when starting in the disjoint condition. The tabular decay rate only showed differences between the stage order for each group, such that starting in the disjoint condition led to overall larger updates of the two-trial back option compared to starting in the conjoint condition. For eligibility, the effect of condition was only found in the decay rate, such that the trace credited to more distant previous choices in the disjoint condition; whereas the strategy weight remained constant across conditions and was higher in the group of participants who started in the disjoint condition. We note that because the experimental conditions and random walk contained systematic deviations in reward magnitude, we chose to make our value function relative (i.e., scaled relative to the last 5 trials, see Methods, S2 Table, S4 and S5 Figs for details and justification) to allow parameter comparison. Therefore, our analysis comparing strategy weights across conditions reflects scaling parameters that are relative to the participants' recent history of experienced value. Despite this potential caveat, model-fitting metrics remained consistent regardless of scaling the value function, confirming the higher performance of the tabular model in the disjoint condition.

Despite the differences in information uncertainty induced by our manipulation, the performance of the eligibility trace strategy remained stable across conditions, which may highlight its suitability across varying uncertainty. The eligibility trace mechanism has shown promise in a variety of tasks, but can become a suboptimal approach when considering an experimental task that contains randomly related events in the time horizon [10,11,34,35]. Furthermore, the eligibility trace solution might have been warranted due to Tanaka et al.'s [26] partially observable feedback presentation and constant stimulus-outcome association. Indeed, overtime agents would fully observe the reward function rather than rely on cognitively dissociating the feedback. Oftentimes, cues help participants maximize reward over long time horizons rather than only focusing on options that are immediately reinforcing [29,36]. However, in the absence of cues, participants need to rely on other avenues of information. One such possibility is the reward signal itself [32].

An unexpected but intriguing finding is the order effects we observed in some of our parameters and model performance indices, indicating overall higher model performance and strategy weights for participants who started in the disjoint condition. One possible interpretation for these order effects is that these participants were primed with less uncertainty by obtaining more information in their first session of the study, which could have led to more accurate responding in the second stage compared to the other group. Human cognition can handle challenges such as sparse rewards, partially

observable states, and long-term consequences, even with limited experience [11,18,37]. However, as the complexity of these environments increases, our understanding of effective strategies to navigate uncertainty remains limited [18]. The belief state must be inferred from observing the environment in which the degree of uncertainty governs the weighing of updates [28,38]. For the group starting in the disjoint condition, this could have reinforced their belief state of the underlying environment regarding the contingencies of immediate and delayed rewards, whereas starting in the conjoint condition might have formed a more uncertain belief state due to incomplete information.

While our study manipulated information uncertainty as a possible driver of strategy use during temporal credit assignment, other factors could contribute to the relative contribution of the two strategies across conditions or stages, such as task difficulty, individual differences, or other learning strategies. It is possible that the differences in strategy use between conditions could have been driven by differences in task difficulty, since the task is harder in the conjoint condition, under low information uncertainty, than in the disjoint condition. While we could not fully isolate the role of task difficulty in our design, our supplemental analyses (S2 Fig) suggest that task difficulty did consistently decrease participants' model fits but did not interact with reward conditions or learning strategies, suggesting it cannot fully explain our condition effects on strategy use. Future experiments may consider using variable delays to help separate the role of information uncertainty, temporal uncertainty, and task difficulty. Furthermore, individuals might differ in a variety of strategies, and one might gain more insight from individual-best-fit, rather than one-fits-all modeling. Capturing a single RL strategy that is shared between individuals can be tricky, especially when considering the vastness of human cognition and potential alternatives [12,39]. For example, perseveration appears to be a vital heuristic for either reducing or eliminating the credit assignment problem and may be adaptive when presented with long-term temporal uncertainty [40,41]. Therefore, future studies examining learning strategies under temporal uncertainty may benefit from incorporating perseveration, working memory, or learning rate asymmetry [42–44].

In summary, temporal CA is a nontrivial problem and the mechanisms implemented in human solutions remain elusive. Despite the complex and demanding nature of the task, participants were able to overcome shifting rewards in a delayed repeat decision task with intervening events. We manipulated the manner of feedback to incentivize participants to increase prospective processes in the hope of characterizing differential strategy use related to information uncertainty. In this regard, we found evidence of increased reliance on a prospective decision strategy that was more efficient when additional information was provided and dependent on the order participants experienced the task. Although a retrospective strategy may be less efficient, its performance appears stable across conditions and could be a generalizable solution. We hope that further investigations will examine new avenues in this experimental design and computational modeling decomposition of individual credit assignment strategies, as well as individual differences in the implementations of these strategies.

## Methods

### Ethics statement

The protocol was approved by the University of Maryland College Park IRB (ID: 1155349–32). Before starting either session, participants completed an online consent form at the start of the experiment and were given free ability to read all parts of the consent form. All participants gave written consent to the full study by selecting 'I agree' as outlined by the approved IRB consent form. They were compensated with a total of $30 for participation and received a bonus dependent on their proportion of selecting the higher valued stimuli.

### Participants

163 participants were recruited from Prolific Academic (https://prolific.com) for an hour and a half long study over two sessions. Prolific inclusion criteria included: fluency in English, ages over 18, and no color blindness. Sessions were separated by two days to one week, but participants were allowed to complete the second session within that flexible interval.

Due to the possibility of external aid, participants were instructed not to use any additional help and were given an end-questionnaire asking if they had used external aid. Although all participants were paid, 13 participants were dropped for not completing the second stage, six were removed for admitting to using external aids, and two were dropped for duplicate stages. The resulting sample contains 142 total subjects (81 males, 60 females, 1 prefer not to say). Ages ranged from 18 to 63 ($M_{age} = 26$, $SD_{age} = 6.58$) with various employment statuses (50 full-time, 34 unemployed, 23 other, 22 part-time, 6 full-time non-paid workers, and 7 missing). See S1 Text for more details.

## Materials

The CA task was built with PsychoPy3 [45]. At the end of all materials, an exit survey was administered with questions assessing whether participants used external aids, the difficulty of the task ("How easy or difficult was the task?") on a sliding-scale of 0 (easy) to 100 (hard), $M_{difficulty} = 60.56$, $SD_{difficulty} = 25.64$, and two open-questions on whether they noticed anything particular or how they had learned the values. We also conducted additional analyses to assess whether task difficulty was a contributing factor to differences between experimental conditions and model performance (S2 Fig).

**CA task.** Participants were given explicit instructions, leading questions, and diagrams of the task structure. The leading questions were aimed at helping participants understand and were intentionally thought-provoking. On each trial (Fig 1A and 1B), participants were instructed to use the mouse to click one of the two objects displayed. They had a total of 15 seconds to make a choice, otherwise no selection was made ($M_{response\ rate} = 99.5\%$). There was a total of 8 objects (4 associated with immediate and 4 with delayed feedback) with 336 trials presenting every unique pair of the 8 objects repeated 12 times. Sixteen images were randomly assigned without replacement to one of the unique objects across the two sessions, resulting in sixteen different object stimuli. Upon selection, a green box surrounded the selected object, and the participant proceeded to the feedback stage. The feedback was dependent on both the reward (starting rewards: -4, -1, 1, 4) and a fixed delay (0, 2). The reward changed over time with three fixed Gaussian random walks, $N(0, .25)$, which were then later rounded to a nearest integer (Fig 1C). This was done to force continuous learning throughout the experiment, as piloting showed quick learning in the beginning trials. At the start of each session, each participant was randomly assigned to one of the three random walks (due to limitations of the online software), meaning that some participants experienced the same random walk across both conditions (28.17%), while the majority experienced different random walks.

**Conditions.** Participants started in either the disjoint (n = 74) or conjoint (n = 68) condition. The conjoint condition gave participants both rewards together, such as receiving a delayed '4' reward from two trials back and an immediate '1' reward from the current trial, which would then be displayed as 5 (Fig 1A). In contrast, the disjoint condition gave participants a dissociable reward. The feedback displayed two boxes titled 'immediate reward' and 'delayed reward', and consequently, did not sum the reward together (Fig 1B).

## Procedure

Each participant filled out a consent form that described the type of task they would receive compensation for. After completing both sessions, participants were given their bonus. Those who did not complete the second session were paid for the first session but did not receive the bonus payment.

## Reinforcement learning models

**Retrospective strategy: Eligibility trace.** The eligibility trace model (we use the abbreviated version 'Elg' in different graphs) uses the Rescorla-Wagner learning rule to calculate the difference between the summed immediate and delayed reward and expected value [7]. This algorithmic implementation is performed in both conjoint and disjoint conditions.

$$\delta_t = r_t(a) - v_t(a) \tag{4}$$

At the current time point (t), this calculates the prediction error ($\delta$) between the actual reward (r) and estimated value (v). Actions (a) are then updated with a replacing eligibility trace, such that the unchosen actions are discounted.

$$et_t(a_i) = \begin{cases} 1 & \text{if } a_i = a_t \\ \lambda_{elg} et_{t-1}(a_i) & \text{if } a_i \neq a_t \end{cases}$$

(5)

The replacing eligibility trace (et) updates all options (i) using a free decay rate parameter ($\lambda_{elg}$), bounded between 0 and 1, for each action. The current selection is updated with a replacing eligibility trace of 1, so that the current selection has no decay [19]. The replacing trace prevents exploding gradients of the value function, as an eligibility trace that exceeds baseline has the potential for exponentiating to infinity. The value function is then updated for each action.

$$v_{t+1}(a_i) \leftarrow v_t(a_i) + \alpha_{elg} \delta_t et_t(a_i)$$

(6)

The learning rate ($\alpha_{elg}$) is a free parameter that determines the magnitude of the update from the reward prediction error (RPE) and eligibility trace. Thus, the temporal sequence becomes highly meaningful for the valuation of past actions.

**Prospective strategy: Tabular update.** The tabular based method (abbreviated 'Tab') has an explicit representation of the temporal sequence [20,46], allowing the agent to prospectively plan for when the reward should be delivered.

$$\delta_t(d = 0) = r_t(a_t) - Q_t(d = 0, a_t)$$
$$\delta_t(d = 2) = r_t(a_t) - Q_t(d = 2, a_{t-2})$$

(7)

Two RPEs are calculated, one for the immediate reward (d = 0) and the other for the outcome of the two-trials previous choice (d = 2) based on the represented delay (d). The Q-learning function considers both the delay and the action chosen.

$$Q_{t+1}(d, a) \leftarrow \begin{cases} Q_t(d, a) + \alpha_{tab} \lambda_{tab} \delta_t(d) & \text{if } d = 2 \\ Q_t(d, a) + \alpha_{tab} \delta_t(d) & \text{if } d = 0 \end{cases}$$

(8)

Both RPEs are used to update the immediate choice and the choice from two trials ago. Note that tabular decay ($\lambda_{tab}$) is only used when we consider delayed feedback (d = 2), whereas immediate feedback is not discounted at all. Because the instructions were explicit about not crediting the action chosen one trial ago, the tabular model skips updates for the one-trial delay. In the conjoint condition, the first prediction error is calculated as the summed reward minus the current selection's value estimate, while the second prediction error is the difference between the same summed reward and the value estimate of the delayed selection from two trials back. This is akin to an eligibility trace that only updates the two signal states (T and T-2). In the disjoint condition, however, the tabular model makes use of the full explicit feedback and performs the same sort of double update. This double update uses the expanded reward function to calculate the two prediction errors, whereby the immediate reward is compared to the current selection and the delayed reward is compared to the selection two trials back.

These credit assignment mechanisms lead to distinct signatures in the model (Figs 3 and S1). Given our experimental design, this leads to a smoother gradient of update for tabular compared to eligibility, particularly in the disjoint condition (S3 Fig). We also attempted to fit other versions of our models to provide more clarity for our model selection (S1 Table).

**Decision rule and hybrid model.** Both models' value functions, eligibility (elg) and tabular (tab), are placed into a SoftMax function. For the independent models, there is a single strategy weight; whereas the hybrid model infuses these

two strategies at decision time through two free parameters which reflect a weighing the strategy contribution at decision time, referred to as strategy weight ($\beta_{elg}$ and $\beta_{tab}$). These betas are used to weigh the value function calculated for each single-strategy model.

$$\pi_t(a) = \frac{exp\left[\beta_{elg}z\left(v_t(a)\right) + \beta_{tab}z\left(\sum Q_t(d,a)\right)\right]}{\sum_{i=0}^{n} exp\left[\beta_{elg}z\left(v_t(a_i)\right) + \beta_{tab}z\left(\sum Q_t(d,a_i)\right)\right]}$$

(9)

When considered alone, the hybrid model can be reduced to either the eligibility or tabular model through a strategy weight of zero assigned to the other strategy. Finally, for all three models (tabular, eligibility, and hybrid), the value function was locally smoothed, by z-scoring action values prior to the current trial before SoftMax decision choice. We chose a 5-trial window to smooth around the lower end of memory recall, and given that adaptive value normalization can handle differing condition contexts [47,48]. By normalizing the value function at decision time, the independent and hybrid models can be compared despite different value scales of the random walks and reward conditions (S4 Fig). This transformation handles statistical obstacles and allows a relative comparison between model parameters. We tested alternative normalization trial windows, which led to similar results in AIC (S2 Table) as well as non-significant changes in parameter distributions (S5 Fig). Specifically, our finding of reduced tabular use in the conjoint condition was independent of the normalization scheme used (S2 Table). As for parameter distributions, outside of expected differences in weight parameters when no scaling is applied, all scaling methods yielded similar parameter values (greatest difference found between 1-trial and 336-trial, p = .1).

**Model behavioral simulations.**   In our stay-switch signature (equation 1), both eligibility and tabular used the participants' best fitting parameters to generate model-predicted choice. Due to low inverse temperature parameters, selection could diverge between iterations and so we repeated the choice-generating process 25 times for each model and each participants' experimental pairs and reward trajectory before fitting the logistic regression. We then compiled the 25 iterations of separate model choices along with participants' actual choice. The significance value of the interaction provides evidence claims on deviation from chance (.5) rather than comparisons between the three models (Fig 3C); nevertheless, the interaction plot with confidence intervals (Fig 3B) inform us of differences between models, as well as similarities and differences between each model's predictions and participants' data by looking at the confidence interval overlaps. In our second posterior predictive check, we also utilized the 25-iteration dataset, which uses participants' best-fitting parameters to generate data (S6 Fig), and found the same results as Fig 4D.

**Model-fitting procedure.**  Each of the three models (eligibility, tabular, and hybrid) was optimized separately for each of the experimental conditions (conjoint, disjoint) using an evolutionary strategy (R package 'DEoptim') [49]. We applied a small penalization prior for each individual to punish extreme values using a gamma distribution ($\alpha = 1.25$, $\theta = 1$) for strategy weight and a beta distribution ($\alpha = 1.25$, $\beta = 1.25$) for learning and decay rates. Pseudo-$R^2$ was calculated by subtracting one from the ratio of the fitted model's negative log-likelihood to that of the null (or random) model for each participant and condition. Additionally, we performed comparisons between AIC and BIC for model comparison and chose AIC given better model recovery (S7 Fig).

**Parameter analyses.**  We first examined the bivariate correlations between the parameters (S3 Table), as well as individual parameter distributions (S8 Fig) partitioned on condition. Although these parameters are dependent on the model they are yoked to and violate independence assumptions of traditional statistics, we reasoned that comparing parameters across condition and stage order group (order of conditions) would still provide additional information to corroborate our hypothesis that tabular and eligibility weights should vary with information uncertainty and to assess whether this effect was impacted by condition order. Due to parameter recovery (S1 Text), we decided to rely on the parameters from our independent models since those were better recovered (S9 Fig), although the differences in parameters from the hybrid model were similar.

We first performed t-tests for a quick intuition on parameter differences between the conjoint and disjoint conditions across all participants (Table 3), then to validate those effects with robust statistical analysis and examine their potential interaction with stage order groups, we ran a multilevel linear regression model predicting each parameter value from condition, stage order group, and their interaction (equation 3). For each of the multilevel models, the first measure of interest was the intraclass correlation (ICC). Typically, the ICC is calculated from an unconditional model which includes only a fixed intercept and random participant intercepts. This model helps to estimate the variance attributable to differences between participants, thereby assessing the reliability of measurements within the same participant across different conditions. In terms of this experiment, this coefficient would represent the correlation or consistency between the participant's conditions. Next, we compared models with and without an interaction between condition and stage order to test whether adding an interaction improved model fit. To do so, a likelihood ratio test using the 'anova' function in R was used, which follows a chi-square distribution. The final statistical test used the interaction in the 'mixed' function, which we note should caution the interpretation of main effects. Lastly, the learning rate of the tabular model required more complex variance and correlation structures for its random effects, known as compound symmetry, where the 'nlme' package was used instead of 'lme4'. This often happens when there is a negative correlation in the parameter between the two conditions that the participant is measured in.

## Supporting information

**S1 Fig. Model credit assignment signatures.** We display our two models (eligibility and tabular) credit assignment mechanism using participants' best fitting parameters to simulate the following value calculation. For each trial (starting on trial 4), we calculated the credit difference, which corresponds to the value update between the current and the next trial for the current trial selection (Time 1). Credit difference values for Time 2–4 represent the same value update calculation, but for the options selected in trials t-1, t-2, and t-3 respectively. We filtered out selection windows with repeated choices to identify the gradient of credit assignment to the last few selections. Values are displayed as a function of prediction error valence (whether the outcome of the current trial is positive or negative), condition (conjoint or disjoint), and model (eligibility or tabular). These simulations align with the prediction of main text Fig 2C, demonstrating the diminishing gradient in credit assignment for the eligibility trace model and the 'jump' or discard of systematically irrelevant delays (t-1 and t-3) for the tabular model.
(EPS)

**S2 Fig. Task difficulty. (A)** Perceived task difficulty ratings collected at the end of the study were not significantly different between our two groups of participants (those who started with the conjoint condition vs those who started with the disjoint condition, $t(140) = -0.12$, $p = .9$). **(B-C)** Effect of task difficulty on model performance. We used negative log-likelihood as a metric of predictive accuracy of our model to participant data, with higher values indicating lower predictive accuracy. **(B)** Using linear regression predicting negative log-likelihood from perceived difficulty, model, and reward condition showed no three-way interaction ($p = .63$), nor any two-way interactions (all $p > .11$). There were significant main effects of perceived difficulty, $t(564) = 2.31$, $p = .02$ and model, $t(564) = 2.13$, $p = .03$, but no effect of reward condition, $t(564) = -1.7$, $p = .09$. This suggests that participants' choices are less consistent with either model as perceived task difficulty increases, rather than difficulty selectively impacting the tendency of participants to rely on one model over the other. (C). A similar analysis was run with the average rate of reward change per participant as a proxy for difficulty (rather than the self-reported perceived difficulty). Reward change was calculated as 0 when the selected object's reward value did not change from the last time it was previously selected and 1 otherwise, with the assumption that higher reward change rate is associated with higher task difficulty. We fit a linear regression predicting negative log-likelihood values from an interaction between model, reward change rate, and reward condition. Once again, there was no three-way interaction ($p = .7$), nor any two-way interactions (all $p > .07$).

There were significant main effects of reward change rate, $t(564) = 9.39$, $p < .001$, model, $t(564) = 2.28$, $p = .02$, and reward condition, $t(564) = -2.74$, $p = .006$. Thus, although difficulty does appear to negatively impact model fit overall, this effect did not vary according to the model or the reward condition.
(EPS)

**S3 Fig. Value function evolution over time.** This figure illustrates the evolution of value functions over time for three representative participants (one for each random walk, labeled 1, 2, and 3 in the facet title) during the presentation of the same experienced stimulus pair trajectory for each participant. The analysis compares the Eligibility Trace (Elg) and Tabular (Tab) learning models under identical conditions: fixed learning parameters ($\alpha = 0.2$, $\beta = 1$, $\lambda = 0.8$) across both information uncertainty manipulations (conjoint vs. disjoint conditions). Each panel displays value changes throughout the experimental session, highlighting the distinct learning trajectories and convergence patterns between the two computational models. Tabular shows smoother learning curves, especially in the disjoint condition, while eligibility has more noise in the value function with more erratic changes and credit jumps.
(EPS)

**S4 Fig. Comparison of value difference heterogeneity between scaling methods.** This figure compares the heterogeneity of model-predicted value differences, calculated as the variance across participants, between experimental conditions, separately for the two scaling approaches (None; 5-Trial Z-Score) and the two models (Eligibility; Tabular). Levene's test of heterogeneity shows that without scaling, there is significant heterogeneity in value difference for both eligibility, $F(5, 278) = 4.37$, $p < .001$, $\eta^2_p = .07$, and tabular, $F(5, 278) = 19.58$, $p < .001$, $\eta^2_p = .19$. However, scaling shows a reduction in heterogeneity for eligibility, $F(5, 278) = 2.63$, $p = .02$, $\eta^2_p = .05$, and a reduction in heterogeneity for tabular, $F(5, 278) = 1.04$, $p = .4$, $\eta^2_p = .02$. The difference between scaling types was significant for eligibility, $F(5, 278) = 4.26$, $p < .001$ and tabular, $F(5, 278) = 19.8$, $p < .001$.
(EPS)

**S5 Fig. Comparison of Z-trial dynamic value scaling.** Depicted are density plots for the distribution of each model's parameters, colored by scaling method. To test for a potential effect of the scaling method, we performed a linear regression predicting parameter value from the interaction between scaling method, parameter, and reward condition. We use the 'emmeans' package to perform pairwise comparisons between the scaling methods. The only significant differences were found between no-scaling and all other dynamic Z-trial methods, specifically: with 1-trial, $b = .1$, $SE = .014$, $t(10152) = 7.27$, $p < .001$; 3-trial, $b = .107$, $SE = .014$, $t(10152) = 7.76$, $p < .001$; 5-trial, $b = .108$, $SE = .014$, $t(10152) = 7.84$, $p < .001$; 7-trial, $b = .11$, $SE = .014$, $t(10152) = 7.96$, $p < .001$; and 336-trial, $b = .136$, $SE = .014$, $t(10152) = 7.96$, $p < .001$. The 336-trial dynamic Z-scaling was the most different from the other scaling methods but did not reach significance: for 1-trial, $p = .1$, 3-trial, $p = .29$, 5-trial, $p = .34$, or 7-trial, $p = .411$.
(EPS)

**S6 Fig. Participants' best fitting parameters for model predicted choice.** Choice data was generated using each of the models with participants' best fitting parameters, with 25 iterations. Those model-generated probabilities were then used to predict participants' actual choices using Equation 2. In other words, this analysis is the same as main text Fig 4D, except model-generated data was simulated using participants' best fitting parameters, instead of parameters sampled from a pre-set distribution.
(EPS)

**S7 Fig. Cross model recovery.** We generated 25 iterations of each model's choice from the 142 participants' two conditions and experienced pair and reward trajectories, then refit them with each model's best fitting parameters. Percentages

show how often each model comparison metric (Akaike information criterion, AIC and Bayesian information criterion, BIC) recovers the true generated model, calculated as the proportion of successful recoveries across all simulations (25 iterations * 142 participants * 2 conditions). Due to better recovery of the hybrid model, we chose to continue model comparison using AIC.
(EPS)

**S8 Fig. Distribution of model parameters.** The distribution of participants' best-fitting learning rate (alpha), inverse temperature weight (beta), and decay rate (lambda) estimated from the eligibility (elg) and tabular (tab) independent models.
(EPS)

**S9 Fig. Parameter recovery (N = 300) for all models, parameters, and conditions.** Parameter recovery was performed by simulating 300 choice datasets from random sets of parameter values, fitting those simulated datasets with the three models (Eligibility, top; Tabular, middle; Hybrid, bottom), and plotting the recovered parameter values as a function of the real parameter values used to generate the data for disjoint (dark gray) and conjoint (light gray) conditions. The closer the locally weighted scatterplot smoothing line (blue-solid) is to the reference line (red-dotted), the better the fit of the generated parameters.
(EPS)

**S1 Table. Additional model comparisons.** Six additional models were tested in relation to our hybrid model using Akaike information criterion (AIC) as our model comparison metric: (i) an eligibility-tabular hybrid with a shared learning rate (Hybrid-1L) between both models; (ii) an eligibility-tabular hybrid with a shared learning rate and shared decay rate (Hybrid-1D) between both models; (iii) an eligibility model with fixed learning and decay rates (Eligibility-Fix) using the mean best fitting parameters from the eligibility model; (iv) a tabular model with fixed learning and decay rates (Tabular-Fix) using the mean best fitting parameters from the tabular model; and (v) a fixed hybrid model (Hybrid-Fix), using the mean best fitting parameters from the hybrid model. We compared all models using a paired Wilcoxon signed-rank tests to the best-fitting (indicated by -) model in each condition. Models that did not significantly differ from the best-fitting model (bolded) represent statistically equivalent alternatives. However, most shared parameter models (1L and 1D) showed similar results to the full hybrid model.
(DOCX)

**S2 Table. Akaike information criterion (AIC) table comparing value normalization schemes.** To test whether our main conclusion is held under different normalization schemes, we compared our theoretically motivated 5-trial scaling window with other scaling schemes (no scaling, 1-, 3- and 7-trial scaling window, and scaling across the entire task). We display AIC values for conjoint and disjoint conditions under two preprocessing approaches: raw value functions (no scaling) versus normalized value functions (5-trial Z-score scaling). The comparison demonstrates that value scaling does not alter model fit patterns, as confirmed by Wilcoxon signed-rank tests comparing conjoint and disjoint conditions for each model and each scaling approach.
(DOCX)

**S3 Table. Correlation between independent model parameters.** Pearson's correlations with diagonals as correlations between disjoint-conjoint, below are conjoint-conjoint correlations and above are disjoint-disjoint correlations. * $p < .05$, ** $p < .01$.
(DOCX)

**S1 Text. Participant demographics, instruction questions, parameter recovery, and RL parameter correlations.**
(DOCX)

## Acknowledgments

A special thanks to Catherine Hartley and Gail Rosenbaum for their invaluable support and guidance during the initial development of this topic and methods.

## Author contributions

**Conceptualization:** Sean R. Maulhardt, Alec Solway.

**Data curation:** Sean R. Maulhardt.

**Formal analysis:** Sean R. Maulhardt, Alec Solway, Caroline J. Charpentier.

**Funding acquisition:** Alec Solway, Caroline J. Charpentier.

**Investigation:** Sean R. Maulhardt, Alec Solway, Caroline J. Charpentier.

**Methodology:** Sean R. Maulhardt, Alec Solway, Caroline J. Charpentier.

**Supervision:** Alec Solway, Caroline J. Charpentier.

**Validation:** Sean R. Maulhardt.

**Visualization:** Sean R. Maulhardt.

**Writing – original draft:** Sean R. Maulhardt, Caroline J. Charpentier.

**Writing – review & editing:** Sean R. Maulhardt, Caroline J. Charpentier.

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
