## [Decision Letter · Decision Letter 0]

26 Mar 2025

Temporal uncertainty influences learning strategy from sequentially delayed rewards

PLOS Computational Biology

Dear Dr. Maulhardt,

Thank you for submitting your manuscript to PLOS Computational Biology. After careful consideration, we feel that it has merit but does not fully meet PLOS Computational Biology's publication criteria as it currently stands. Therefore, we invite you to submit a revised version of the manuscript that addresses the points raised during the review process.

Please submit your revised manuscript within 60 days May 26 2025 11:59PM. If you will need more time than this to complete your revisions, please reply to this message or contact the journal office at ploscompbiol@plos.org. Please include the following items when submitting your revised manuscript:

We look forward to receiving your revised manuscript.

Kind regards,

Alireza Soltani

Academic Editor

PLOS Computational Biology

Tobias Bollenbach

Section Editor

PLOS Computational Biology

**Journal Requirements:**

Potential Copyright Issues:

i) Figures 1A, and 1B. Please confirm whether you drew the images / clip-art within the figure panels by hand. If you did not draw the images, please provide (a) a link to the source of the images or icons and their license / terms of use; or (b) written permission from the copyright holder to publish the images or icons under our CC BY 4.0 license. Alternatively, you may replace the images with open source alternatives. See these open source resources you may use to replace images / clip-art:

6) Please ensure that the funders and grant numbers match between the Financial Disclosure field and the Funding Information tab in your submission form. Note that the funders must be provided in the same order in both places as well. Currently, "University of Maryland start-up funds" is missing from the Funding Information tab.

**Reviewers' comments:**

Reviewer's Responses to Questions

Reviewer #1: The paper by Maulhardt, Solway, and Charpentier examined how humans solve the temporal credit assignment when they receive both immediate and delayed rewards. Using computational modeling, the authors found that when participants have no explicit knowledge about the source of each part of the rewards, they used a retrospective algorithm based on eligibility traces to learn the values of stimuli, which leads to rewards being associated with recent actions that did not causally lead to the current reward. However, when they know the values of immediate and delayed rewards respectively, they instead use a prospective tabular algorithm which more accurately associates reward with the correct action. The task design is novel but straightforward and provides important new insight about human reinforcement learning. To enhance the clarity and impact of the results, the modeling approach could benefit from some more elaboration. Some questions and proposed additional analyses are presented below that could strengthen the results and lend more support the current conclusions.

1. In the result of Fig. 3, it is unclear whether the models were simulated using each subject’s best-fitting parameters or parameters drawn from some prior distribution. In order to validate these models, it would be best to fit them to behavior first and then simulate using the best-fitting parameters. This applies similarly to Fig. 4.

2. Some panels in Fig. 3 are not referenced in the main text and could be moved to the supplementary materials to highlight the main behavioral signature of interest. While the effect of reward on stay probability in each combination of condition and time is the quantity of focus here, it is not directly visible from the estimated marginal means. It would be ideal if this effect could be summarized with a single number for each condition and time, maybe by treating reward as a single continuous predictor and looking at the regression coefficient under different conditions and trial times.

3. Looking at the example pairs in Fig 4 is a solid approach but does not lead to a lot of additional insights. It is not surprising that both participants and the model could learn to change their behavior when reward contingencies change. Can this be used as evidence for or against either type of models?

4. In Fig 4C, the fixed effect of disjoint * eligibility is not significant, contradicting the figure caption.

5. In the conjoint condition, do the estimated values used to calculate prediction error (Eq. 4 and 7) correspond to the sums of the current action and action two trials back? This seems more sensible, since participants should know that the reward is the sum of immediate and delayed rewards. The equations seem to suggest that the prediction error is calculated only using the current action (Eq. 4) or separately for the current and previous action (Eq. 7).

6. In the disjoint condition, it seem plausible that tabular-based learners (or maybe even eligibility trace-based learners) would calculate two separate prediction errors using the immediate and delayed rewards, which doesn’t seem to be what Equation 7 suggest. How is this taken into account by the models?

7. Is z-scoring the values crucial for the results? It seems like an unconventional choice. How would the results change without it?

8. The model comparison results could be corroborated with more careful analyses and interpretation. In Table 2, having AIC, BIC, and pseudo-R^2 for model comparison feels redundant. When comparing the hybrid model with the two other models, pseudo-R^2 is not very useful since it does not take into account parameter count. Furthermore, the results of AIC and BIC contradict each other in three of the four conditions, which makes the result inconclusive. The authors may wish to perform model recovery using the two metrics to identify which metric leads to better recovery of the true model. The comparisons between eligibility and tabular models are not affected as much since they have the same number of parameters, but it’s hard to tell the effect size of their differences without knowing the standard error of these metrics. Bayesian model selection could also be useful.

9. Based on Table 2, conjoint conditions are better fit by eligibility models, but disjoint conditions (especially stage 2) are not necessarily better fit by tabular models. This contradicts the heading of the previous section, and seemingly also the result in Fig. 4C. Will the results change if the models were simulated using participants’ best-fitting parameters?

Reviewer #2: Maulhardt et al. investigated how humans engage in temporal credit assignment. Using a modified two-armed bandit task, the authors have shown that humans uses a hybrid of eligibility trace and tabular update, the weight of which is influenced by the degree of information uncertainty. The authors have used a variety of analyses to demonstrate that participants are able to perform this temporal credit assignment problem and that the posterior predictive check of different models matches participants behavior under joint/disjoint conditions.

Major points:

1. Despite the importance of the research topic and solid data analysis, I am not sure if the current set of experiments and analysis convinces me that the eligibility trace and tabular update models are the most suitable model candidates to account for human’s decision making in this temporal credit assignment task. This is mainly because the authors have not tested other competing models (e.g., single step TD learning) or reduced version of the two main models (e.g., fix some parameters to a fixed value) to justify that all the parameters are necessary. The overall inconclusive results on the model comparison (AIC, BIC) also makes me wonder whether splitting people based on their individual’s best fit models and examining their behavior could provide more insights into how the usage of different algorithms is related to different behavioral patterns.

2. I am also not totally convinced by the argument of temporal uncertainty influences the strategy adopted during temporal credit assignment. Why it is not just task difficulty? If the argument is about task difficulty, let’s think about another hypothetical experiment where delayed reward is 3 steps into the future. Since the task is harder (requires to hold more distant items in working memory), I would expect that participants put more weight on the eligibility trace algorithm, while in this case the information uncertainty I assume is the same as 2-step situation since the immediate and delayed reward is always separated. I would suggest the authors either conduct more experiments that manipulate information uncertainty in different ways to provide more support for this argument, or maybe be more careful and clearer when defining what they mean by uncertainty.

Other comments:

1. The authors should cite Bruinsma, S., Petzschner, F., & Nassar, M. Perseverative Behavioral Sequences Aid Long-Term Credit Assignment. (A CCN proceeding)

2. On Figure2, I assume O1 means Observation 1? It can be easily mistaken to be 0. I would recommend either write out a longer abbreviation (obs?) or make it more clearer about what S and O stands for in the caption.

3. On page 10, the authors said that ‘we selected sequences where participants chose a delayed option and it reappeared as a potential choice three trials later’ - it would be helpful to know how many trials (the proportion of the total trials) this subset refer to.

4. On page 11, I am not sure I understand the interpretation of the three-way interaction disjoint*+*2F. What is the baseline level that this interaction term is compared to? To get a better idea of what a three way interaction is, it may be helpful to split the data based on one of the factor and redo two-way interaction to examine the effect more directly.

5. On page 13, the authors claimed that their priorr parameter selection ‘maintained consistent parameters across the two models’ - how is this achieved?

6. For Figure4, it seems that for all the pairs, the initial better option will in the end drift away to be the inferior option - is this by design in the pseudo-random random walk schedule? I understand that the restless aspect of the task is to make sure that the participants keep learning, but why introduce this structural drift early vs. late task?

7. What is the overall distribution histogram of the weight parameter in the hybrid model?

**Have the authors made all data and (if applicable) computational code underlying the findings in their manuscript fully available?**

Reviewer #1: None

Reviewer #2: Yes

PLOS authors have the option to publish the peer review history of their article (what does this mean? ). If published, this will include your full peer review and any attached files.

**Do you want your identity to be public for this peer review?** For information about this choice, including consent withdrawal, please see our Privacy Policy .

Reviewer #1: No

Reviewer #2: No

**Figure resubmission:**
---

## [Decision Letter · Decision Letter 1]

22 Jul 2025

PCOMPBIOL-D-25-00171R1

Information uncertainty influences learning strategy from sequentially delayed rewards

PLOS Computational Biology

Dear Dr. Maulhardt,

Thank you for submitting your manuscript to PLOS Computational Biology. After careful consideration, we feel that it has merit but does not fully meet PLOS Computational Biology's publication criteria as it currently stands. Therefore, we invite you to submit a revised version of the manuscript that addresses the points raised during the review process.

Please submit your revised manuscript within 30 days Sep 21 2025 11:59PM. If you will need more time than this to complete your revisions, please reply to this message or contact the journal office at ploscompbiol@plos.org. Please include the following items when submitting your revised manuscript:

We look forward to receiving your revised manuscript.

Kind regards,

Alireza Soltani

Academic Editor

PLOS Computational Biology

Tobias Bollenbach

Section Editor

PLOS Computational Biology

**Reviewers' comments:**

Reviewer's Responses to Questions

Reviewer #1: I appreciate the author’s thorough effort in the revision which has substantially improved the manuscript. I have a few follow-up comments:

1. The authors have simplified and clarified Fig. 2, but I still find it somewhat hard to interpret. It would be great if we can see qualitatively distinct predictions made by the two models (similar to Fig. 1B in Witkowski et al., 2025, and Fig. 2E-F in Jocham et al., 2016), although the quantitative comparisons are certainly valid and consistent with the rest of the paper. I believe the eligibility model predicts that positive/negative outcomes at trial T lead to more/less likelihood to repeat choices made in trial T, T-1, T-2, T-3, etc. with decreasing effect. The tabular model predicts that reward outcomes at trial T gets attributed only to choices in trial T and T-2, and nothing else. This should be especially apparent if we look at the simulations, but it is hard to discern currently. I believe looking at the difference between P(stay|+ outcome) and P(stay|- outcome) would reflect win-stay lose-switch more clearly than just looking at the marginal means individually.

2. For model recovery, a confusion matrix similar to the one used in Box 5 of Wilson & Collins (2019) would be much more informative than the current S2 table. Due to the importance of parameter values to the conclusion of the study, adding a parameter recovery analysis would be valuable and should not take excessive effort given the authors have already performed model recovery.

3. I appreciate the author’s clarification on the z-scoring approach. However, my concern is that it is adding something post-hoc into a model that is supposed to reflect cognitive mechanisms. Participants may or may not adapt their strategies to different magnitudes of rewards, but z-scoring based on previous five trials seems to be an arbitrary decision. The absolute scale of the reward could definitely matter within a single session, which the rolling z-scoring approach interferes with. Why not normalize the rewards based on the scale of the entire reward trajectory, or scale the betas based on the range of rewards? Whether or not z-scoring is applied shouldn’t affect model comparison (since the same trajectory should be fitted to by all models). If the concern is with comparing the model parameter across conditions, there should be statistical techniques that work for unbalanced sample sizes and the random walk condition can be included as a nuisance variable. As the author mentioned, despite the z-scoring procedure changing the beta weights significantly, this choice does not affect the parameter comparison conclusions qualitatively. I am still not convinced that the rolling z-scoring is necessary or justified and suggest the authors focus on results with either no z-scoring or other ways to remove the effect of reward scales.

4. For model comparison, it is necessary to include some information on the distribution of AICs to see if the differences in fit are actually significant. For example, this could be done with a rank-sum test or Bayesian Model Selection (Rigoux et al., 2014).

Reviewer #2: The manuscript has been largely improved. However I still have a couple of lingering questions and think they are important to address before the acceptance of the paper.

Major point:

(1) I am a bit concerned/confused by the value normalization after reading the response from the authors to reviewer 1. Why the value normalization leads to a change in # of subjects? The authors also argue that most of existing work does not have changed reward magnitude - which is not 100% true - there are a good amount of work in bandit task that generate reward from a gaussian distribution. I do agree with the authors that those work do not necessarily seek a cross-condition comparison though. I guess my confusion here is: what exactly does value normalization change? The authors have mentioned some possible scenarios, but I could use some help of some more concrete examples. Also, do the results from the current paper change drastically if value normalization is not applied?

(2) hybrid model:I understand that the authors try to make the case that given different experiment conditions - that is, different information uncertainty - people may adopt different exploration strategy (or put different weights on a set of strategies). In that case, would it be more intuitive in data analysis to compute weight as the ratio of beta for two strategies and see how that changes across conditions? Relatedly, on an implementation level, are these two betas fit once, and then for the single-strategy models the authors use one of the fitted beta and fix the other one to be 0? Or do they do three separate fittings for three models?

Minor points:

(1) For model recovery (Table S2), how many total simulations was run?

(2) when interpreting model comparison results, the authors refer to delta AIC = 2 as a threshold for meaningful difference. Is this criterion obtained from any book/paper citation?

(3) on p6: 'This leads to our final questions: whether reward learning under temporal uncertainty can be empirically solved by a mixture of prospective and retrospective strategies' <- Should this be information uncertainty given the updated terminology?

(4) the result section when describing Fig 3B: would be helpful if the authors can remind the readers that they are using the overlapped confidence intervals as an indicator for model performance - and maybe report the CI stats for some important comparisons. the current language (e.g., ‘follow participants behavior’, ‘had mixed patterns’) is a little vague for the discussion section.

(5) I appreciate the additional analysis related to task difficulty (Fig S1). However I am pretty confused by what (B) and (C) means. Why use AIC as the DV? for (B), in caption it suggests negative coef for IVs, but on the graph it seems that the lines have positive slope? In the caption the authors mentioned ‘make people more stochasticity and less consistent’ - is this based on interpreting smaller AIC or something else?

**Have the authors made all data and (if applicable) computational code underlying the findings in their manuscript fully available?**

Reviewer #1: None

Reviewer #2: Yes

PLOS authors have the option to publish the peer review history of their article (what does this mean? ). If published, this will include your full peer review and any attached files.

**Do you want your identity to be public for this peer review?** For information about this choice, including consent withdrawal, please see our Privacy Policy .

Reviewer #1: No

Reviewer #2: No

**Figure resubmission:**
---

## [Decision Letter · Decision Letter 2]

31 Oct 2025

PCOMPBIOL-D-25-00171R2

Information uncertainty influences learning strategy from sequentially delayed rewards

PLOS Computational Biology

Dear Dr. Maulhardt,

Thank you for submitting your manuscript to PLOS Computational Biology. After careful consideration, we feel that it has merit but does not fully meet PLOS Computational Biology's publication criteria as it currently stands. Based on the reviewers’ most recent comments, we would like to ask you to submit a revised version of your manuscript that addresses their feedback. **Please note that your revision will not be sent out for external review but will be evaluated at the editorial level.**

Please submit your revised manuscript within 30 days Dec 31 2025 11:59PM. If you will need more time than this to complete your revisions, please reply to this message or contact the journal office at ploscompbiol@plos.org. Please include the following items when submitting your revised manuscript:

We look forward to receiving your revised manuscript.

Kind regards,

Alireza Soltani

Academic Editor

PLOS Computational Biology

Tobias Bollenbach

Section Editor

PLOS Computational Biology

**Reviewers' comments:**

Reviewer's Responses to Questions

**Comments to the Authors:**

Reviewer #1: I really appreciate the thorough analyses conducted in the revision, especially in model recovery and comparison, as well as the authors’ effort in clarifying points of confusion. I believe most of my concerns have been addressed. Here are a few final lingering questions:

1. I’m confused at why the eligibility and tabular models display such similar behavior in the WSLS adherence plot. As shown in the illustration in Figure 2C, these models should have qualitatively distinct credit assignment behavior. Specifically, how could reward at lag 0 lead the eligibility model to accurately repeat object at lag 2, but not repeat object at lag 1? It makes sense that there is no real systematic relationship between outcomes at 0 and the choice at lag 1 and 3. However, doesn’t the eligibility model predict that participants will nonetheless falsely associate that outcome with those choices, especially choice at lag 1 due to it being temporally closer to the outcome at lag 0? While I think these analyses do not necessarily undermine the author’s claims, the results would be much clearer and stronger if the authors can directly demonstrate the behavior predicted in Fig 2C in the data and in the models.

2. I understand the author’s rationale for normalizing the values, as different random walk reward trajectories will lead to different scales of values, which then leads to differences in the scale of model parameters that may reflect artifacts of model fitting procedure and not psychologically meaningful parameters. It is very reasonable that some normalization is needed to allow for cross-session comparison. However, the choice of dynamic z-scoring by 5 trial windows is still arbitrary, especially compared to a simple global scaling of the entire value trajectory. As participants make different choices that lead to different outcomes, their Q-values will naturally diverge with more trials, and I don’t believe it is necessary to use a dynamic z-scoring procedure to fully remove this heterogeneity. If different normalization strategies lead to different results, then it should be more important to show that the main conclusion hold under different normalization schemes. Looking at reviewer #2’s comment, I also wonder why the authors did not look at the ratio of each strategy’s parameters in the hybrid model. Since both parameters are fit to the same random walk trajectory, wouldn’t looking at their ratio remove the effect of reward scales? If so, wouldn’t comparing such ratios remove the need for any normalization?

Reviewer #2: The authors have addressed all my concerns. Just a quick note that please also remember to update the osf repo to reflect the new analysis that has been done. Thanks for the hard work.

**Have the authors made all data and (if applicable) computational code underlying the findings in their manuscript fully available?**

Reviewer #1: None

Reviewer #2: Yes

PLOS authors have the option to publish the peer review history of their article (what does this mean? ). If published, this will include your full peer review and any attached files.

**Do you want your identity to be public for this peer review?** For information about this choice, including consent withdrawal, please see our Privacy Policy .

Reviewer #1: No

Reviewer #2: No

**Figure resubmission:**
---

## [Editor Report · Decision Letter 3]

29 Dec 2025

Dear Mr. Maulhardt,

We are pleased to inform you that your manuscript 'Information uncertainty influences learning strategy from sequentially delayed rewards' has been provisionally accepted for publication in PLOS Computational Biology.

Best regards,

Alireza Soltani

Academic Editor

PLOS Computational Biology

Tobias Bollenbach

Section Editor

PLOS Computational Biology

---

## [Editor Report · Acceptance letter]

25 Mar 2025

PCOMPBIOL-D-25-00171R3

Information uncertainty influences learning strategy from sequentially delayed rewards

Dear Dr Maulhardt,

I am pleased to inform you that your manuscript has been formally accepted for publication in PLOS Computational Biology. Your manuscript is now with our production department and you will be notified of the publication date in due course.

With kind regards,

Anita Estes
